ecology, environmental science, evolution

pastoralism, millet, ancient DNA, isotope analysis, steppe archaeology

**Authors for correspondence:**
Taylor R. Hermes
e-mail: trhermes@gshdl.uni-kiel.de
Michael D. Frachetti
e-mail: frachetti@wustl.edu
Cheryl A. Makarewicz
e-mail: c.makarewicz@ufg.uni-kiel.de

†Deceased 12 March 2018.

# Early integration of pastoralism and millet cultivation in Bronze Age Eurasia

Taylor R. Hermes[1,2], Michael D. Frachetti[3], Paula N. Doumani Dupuy[2,4], Alexei Mar'yashev[5,†], Almut Nebel[1,6] and Cheryl A. Makarewicz[1,2]

[1]Graduate School 'Human Development in Landscapes', Kiel University, Leibniz Straße 3, 24118 Kiel, Germany
[2]Institute of Prehistoric and Protohistoric Archaeology, Kiel University, Johanna-Mestorf-Straße 2-6, 24118 Kiel, Germany
[3]Department of Anthropology, Washington University in St Louis, One Brookings Drive, St Louis 63130, USA
[4]School of Humanities and Social Sciences, Nazarbayev University, Kabanbay Batyr Avenue 53, Astana 010000, Kazakhstan
[5]Margulan Institute of Archaeology, Dostyk Avenue 44, Almaty 480100, Kazakhstan
[6]Institute of Clinical Molecular Biology, Kiel University, University Hospital Schleswig-Holstein, Rosalind-Franklin Straße 12, 24105, Kiel, Germany

TRH, 0000-0002-8377-468X

Mobile pastoralists are thought to have facilitated the first trans-Eurasian dispersals of domesticated plants during the Early Bronze Age (*ca* 2500–2300 BC). Problematically, the earliest seeds of wheat, barley and millet in Inner Asia were recovered from human mortuary contexts and do not inform on local cultivation or subsistence use, while contemporaneous evidence for the use and management of domesticated livestock in the region remains ambiguous. We analysed mitochondrial DNA and multi-stable isotopic ratios ($\delta^{13}$C, $\delta^{15}$N and $\delta^{18}$O) of faunal remains from key pastoralist sites in the Dzhungar Mountains of southeastern Kazakhstan. At *ca* 2700 BC, Near Eastern domesticated sheep and goat were present at the settlement of Dali, which were also winter foddered with the region's earliest cultivated millet spreading from its centre of domestication in northern China. In the following centuries, millet cultivation and caprine management became increasingly intertwined at the nearby site of Begash. Cattle, on the other hand, received low levels of millet fodder at the sites for millennia. By primarily examining livestock dietary intake, this study reveals that the initial transmission of millet across the mountains of Inner Asia coincided with a substantial connection between pastoralism and plant cultivation, suggesting that pastoralist livestock herding was integral for the westward dispersal of millet from farming societies in China.

## 1. Introduction

Mounting archaeological research in the Eurasian steppes demonstrates that pastoralists associated with diverse Bronze Age cultures and later nomadic empires engaged in farming to a far greater degree than previously thought [1–6]. By *ca* 1600 BC, archaeobotanical evidence links Inner Asian herding communities to the cultivation of southwest Asian domesticates of wheat (*Triticum* spp.) and barley (*Hordeum* spp.), alongside broomcorn millet (*Panicum miliaceum*) and foxtail millet (*Setaria italica*) domesticated in northern China. Around the same time, these key crops were also widely adopted by societies at opposite ends of Eurasia [7–10]. Interestingly, the earliest indication of people exchanging crops through Inner Asia is represented by carbonized seeds of wheat, barley and broomcorn millet in human cremation cists (*ca* 2500–2200 BC) in two small pastoralist settlements, Begash and Tasbas, in the Dzhungar Mountain foothills of southeastern Kazakhstan [5,11,12]. Without direct evidence for local cultivation or subsistence use, the mortuary context of

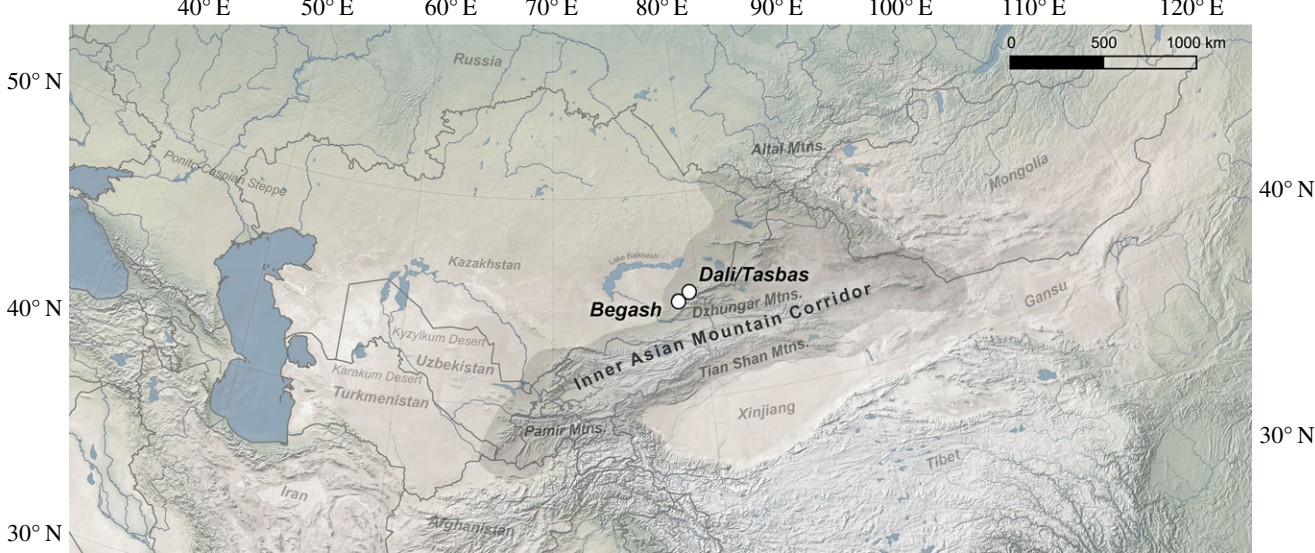

**Figure 1.** Locations of Dali, Tasbas and Begash within the IAMC. (Online version in colour.)

these seeds raises the possibility that domesticated cereals initially spread to Inner Asia as traded goods for symbolic purposes, rather than as staple foods [13]. To date, an untested dispersal model is that crops were cultivated and transmitted as part of strategies tightly linked to herding sheep, goats and cattle. However, the general paucity of faunal remains of domesticated bovids dating to the Early Bronze Age in Inner Asia or northern China obscures the timing of the spread of pastoralist subsistence through this continental crossroad [14].

Here, we examined the early transmission of bovid domesticates of sheep, goat and cattle in the Dzhungar Mountains and the potential integration of pastoralist herding with plant cultivation during the initial Eurasian interchange of plant and animal domesticates. We performed genetic and isotopic analyses of faunal bones and teeth from the newly discovered Early Bronze Age settlement site of Dali dating to *ca* 2700 BC, in addition to faunal remains from younger strata at nearby Begash and Tasbas containing millet remains (*ca* 2345 BC to AD 30; figure 1). These analyses, respectively, provide concrete taxonomic identifications of faunal remains and resolve dietary intake of livestock at multi-year and seasonal scales in order to assess when and how pastoralists facilitated early processes of food globalization in Eurasia.

## 2. Archaeological background

Pastoralism using sheep, goats and cattle has been a core subsistence strategy of Inner Asian societies for arguably more than 5000 years [15]. This mobile lifeway is thought to have spread eastward across the Eurasian steppe during the Eneolithic/Early Bronze Age as part of migrations of Yamnaya herders (3500–2700 BC) along a route from the Pontic-Caspian western steppe to the Altai mountains [16], giving rise to the Afanasievo culture (3100–2500 BC) [17]. Human genomic data showing indistinguishable genetic variation between Early Bronze Age communities of the western steppe and the Altai region support this purported migration [18–21]. However, Neolithic Altaic communities have chronological and cultural overlap with 'Afanasievo' groups [22], for which little is known about food production strategies using purported domesticated plants or animals.

An alternative theory to pastoralism being translocated across the steppe is that pastoralist livestock spread to Inner Asia via limited human mobility and cultural transmissions along foothill ecozones linking the Iranian Plateau, Altai Mountains and western China, known as the Inner Asian Mountain Corridor (IAMC) [23].

Based on the macrobotanical record, one of the most important domesticated plants at human settlements in the central IAMC spanning the Early Bronze Age (2300 BC) to the Mongol period (AD 1200–1400) was broomcorn millet [3,5,12], a unique crop characterized by rapid seeding time and drought tolerance compared to Near Eastern wheat and barley [24]. Nevertheless, the early chronology of local cereal cultivation at quantities that would sustain human subsistence in this region remains hotly debated [1,2,5,11,12,25–27]. Unlike settlements of sedentary communities that generate considerable quantities of food waste in discrete middens, pastoralist occupations are often ephemeral and dispersed and yield little cultural material [15], such as the crop-processing debris that some consider to be unambiguous evidence for local cultivation [25]. Documenting human millet intake expected with the adoption of this isotopically distinct $C_4$ crop using stable carbon isotope analysis has been severely impeded by the paucity of human remains dated to the third millennium BC.

## 3. Material and methods

### (a) Excavation, radiocarbon dating and sample selection

Dali is located at 1500 m.a.s.l. in the Bayan-Zhurek range of the Dzhungar Mountains of Kazakhstan, at the heart of the IAMC and approximately 1 km southwest from Tasbas (figure 1). Previous excavations at Tasbas uncovered a small area dating to the Early Bronze Age (2655–2480 cal BC; electronic supplementary material, table S1), revealing a stone cremation cist containing wheat seeds ($n = 5$) and microliths ($n = 3$); no ceramics and only fragmented and burned animal bones identifiable to genus ($n = 12$) were recovered [11]. At Begash (2345–2080 cal BC; electronic supplementary material, table S1), a similar mortuary practice included seeds of wheat ($n = 13$), barley ($n = 1$) and broomcorn millet ($n = 59$) [5,12], and faunal remains more clearly indicate exploitation of pastoralist livestock based on less fragmented specimens identifiable to genus [28].

Excavations of the Dali settlement were conducted in 2011, 2012, 2014,[1] following established methods [11,29]. New and previously published radiocarbon dates from Dali, Begash and Tasbas were modelled according to occupational layers in OxCal v. 4.3.2 with the INTCAL13 calibration curve [30], providing start and end dates of phases (electronic supplementary material, table S1). Faunal specimens for aDNA and isotope analyses were sampled from securely dated strata.

## (b) DNA analysis

Morphological identifications of domesticated sheep and goat skeletal specimens from sites located in the IAMC are vulnerable to a high degree of taxonomic uncertainty due to pronounced fragmentation of skeletal material and overlapping diagnostic criteria with endemic wild taxa, including argali (*Ovis ammon*), urial (*Ovis vignei*), Siberian ibex (*Capra sibirica*) and markhor (*Capra falconeri*) [31]. Collagen peptide mass fingerprinting (Zooarchaeology by Mass Spectrometry, aka 'ZooMS'), previously used to identify purported early livestock in Central Asia [32] and elsewhere, is unsuitable because of coarse taxonomic resolution limited to genus-level, which does not distinguish between domesticated and wild lineages. As a sensitive method for identifying *Ovis* and *Capra* species, a 110-bp stretch of the mitochondrial cytochrome *b* gene (*MT-CYB*) from suspected sheep and goat remains was amplified using primers CapFC1 (5′-CTCTGTAACTCACATTTGTC-3′) and CapRB1b (5′- GTTTCATGTTTCTAGAAAGGT-3′) [33]. Two independent DNA extractions for each specimen were conducted following established protocols [34]. See electronic supplementary material, text S1 for PCR and analytical methods.

## (c) Stable isotope analysis

Millets use the $C_4$ photosynthetic pathway and have substantially higher $\delta^{13}C$ values by approximately 14‰ relative to $C_3$ plants [35]. Intra-annual records of herbivore dietary intake were recovered through incremental measurement of carbon and oxygen isotopes in tooth enamel bioapatite ($\delta^{13}C_{apa}$ and $\delta^{18}O_{apa}$), which archives the proportion of $C_4$ or $C_3$ plants consumed in whole diet [36] and cycles of seasonal environmental inputs via body water [37], respectively. We analysed second and third mandibular molars, forming during the first and second years of life, respectively [38,39]. Enamel powder was collected from 1 mm wide bands oriented perpendicular to the tooth growth axis on the buccal side from crown to cervix [39,40]. Enamel pre-treatment followed reference [38]. Bioapatite carbonates were analysed at the Leibniz Labor, Kiel University with a Finnigan MAT 253 mass spectrometer coupled to a Kiel IV device using reactions with 100% orthophosphoric acid at 75°C. Duplicates were run for every approximately 10 samples, and two in-house enamel standards were run for every approximately five samples; analytical precision was 0.05‰ for carbon and 0.07‰ for oxygen. Measurements were calibrated using the carbonate standard NBS 19 ($\delta^{13}C = +1.95$‰ and $\delta^{18}O = -2.2$‰) and expressed relative to the international Vienna PeeDee Belemnite (VPDB) standard in delta notation.

We analysed the carbon and nitrogen isotopic composition of bone collagen ($\delta^{13}C_{col}$ and $\delta^{15}N_{col}$), reflecting dietary protein at lifetime scales derived from plant ingesta [41] and nitrogen inputs from farming and pasturing activity [42], respectively. Collagen was extracted following Tuross *et al*. [43] using 0.5 M EDTA with a defatting step for modern samples and mass spectrometry was performed using a EuroVector Euro EA elemental analyser coupled to a GVI IsoPrime in continuous flow mode at the Boston University Stable Isotope Laboratory. Analytical error was 0.1‰ for $\delta^{13}C$ and 0.2‰ for $\delta^{15}N$ using peptone and glycine standards run for every approximately 15 samples. Isotopic values are expressed relative to VPDB for $\delta^{13}C$ and atmospheric nitrogen for $\delta^{15}N$ in standard delta notation. Collagen samples with elemental C : N ratios from 2.9 to 3.6 and also %C and % N within established ranges were accepted for analysis [41]. We also analysed previously published human $\delta^{13}C_{col}$ values from directly radiocarbon dated human individuals across the wider steppe region [27,44–47] (*n* = 174; dataset S5).

## (d) Modelling livestock dietary intake

Statistical analyses were performed in R v. 3.4.1 [48]. We estimated time of year for $\delta^{13}C_{apa}$ values using an approach adapted from Balasse *et al*. [49], which normalizes differences in inter-individual tooth eruption and enamel maturation rates. Over annual rhythms of $\delta^{18}O_{apa}$ values, which represent seasonal cycles of temperature reflected in meteoric water [49] and rainfall amount to a lesser extent [50], we modelled cosine periods as 365 'Julian days' with January 15th as day zero. The strong seasonal climate in Inner Asia imparts regular oscillation in the oxygen isotopic composition of open and leaf water sources reflected in herbivore tooth enamel [38], overriding transient fluxes in environmental and biological systems [37,50]. We used MixSIAR to generate Bayesian estimates of $C_4$ plant intake as dietary percentages [51,52]. We minimized temporal error by calculating means of $\delta^{13}C_{apa}$ sequences by two-month intervals for each tooth within archaeological periods at each site as fixed effects.

Critically, the carbon isotopic composition of local steppe vegetation was estimated from dietary $\delta^{13}C$ values derived from (1) tooth enamel bioapatite of ancient red deer (*Cervus elaphus*), saiga (*Saiga tatarica*) and modern sheep herded in semi-arid lowlands near the sites, (2) bone collagen of ancient red deer, ancient Siberian ibex (*Capra sibirica*), ancient argali (*Ovis ammon*), ancient boar (*Sus scrofa*) and modern steppe tortoise (*Agrionemys horsfieldii*) and (3) summertime-forming hair of modern saiga [53]. These taxa substantially share dietary niches with pastoralist livestock and thus serve to detect wild $C_4$ vegetation in local ecosystems that would confound isolation of millet foddering. Unlike their European counterparts, Central Asian red deer graze throughout the year and seasonally range into arid lowland pastures [54]. Migratory saiga antelope are voracious grazers in deserts and semi-arid steppes [53]. Bones of modern steppe tortoise, which are generalist feeders [55], were collected from the Saryesik-Atryan Desert enclosed by Lake Balkhash (figure 1). Central Asian wild boar consume a broad range of small and medium animals and herbaceous plants, including summer grasses [56]. See electronic supplementary material, text S2 for MixSIAR parameters and diet-tissue spacing factors used to derive dietary $\delta^{13}C$ values from $\delta^{13}C$ values of tissues from various taxa shown in figure 2.

# 4. Results

## (a) The Dali settlement complex

Phase 1 occupation levels at Dali date to 2705–2545 cal BC (electronic supplementary material, figure S1 and table S1) and include pit house architecture not previously described for the region. Ceramics from phase 1 resemble those from Altaic cultures of the third millennium BC (electronic supplementary material, figure S3). A human parietal bone recovered from the pit house exhibits a genetic composition most similar to Eurasian hunter–gatherers, including Eneolithic horse herders at Botai, with approximately 20% admixture from Neolithic populations from the northeastern Iranian plateau, while there was not a contribution of western steppe Yamnaya or Afanasievo ancestry [57]. Phase 2 of Dali dates to 1645–1520 cal BC (electronic supplementary material, figure S2 and table S1) and is characterized by a new cultural

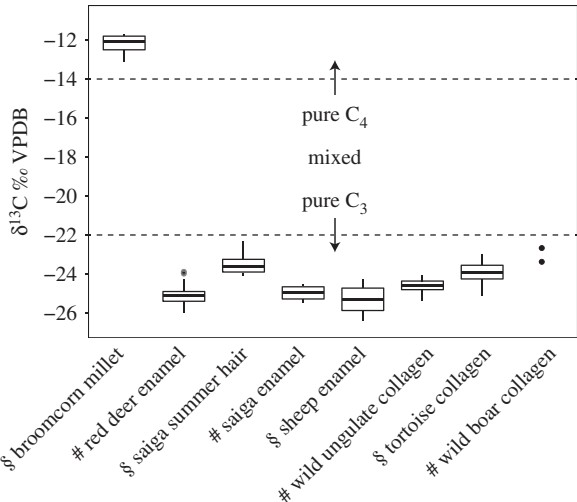

**Figure 2.** $\delta^{13}C$ values of broomcorn millet versus dietary $\delta^{13}C$ values of reference herbivores from study region. Symbol § denotes modern samples, including saiga antelope hair ($n = 22$), sheep tooth enamel ($n = 51$), steppe tortoise bone collagen ($n = 6$). Symbol # denotes ancient samples, including red deer enamel ($n = 49$), saiga tooth enamel ($n = 6$), red deer and ibex bone collagen ($n = 21$) and wild boar bone collagen ($n = 2$). See electronic supplementary material, text S2 for diet-tissue spacing factors and dataset S2 for raw data.

pattern (electronic supplementary material, figure S4). Architecture radically changes during this phase to rectilinear stone foundations, similar to structures recovered at contemporaneous sites in the region, such as nearby Begash [29] and Adunqiaolu in Xinjiang [58]. Human individuals ($n = 3$) excavated from Dali's mortuary complex are contemporaneous with phase 2 and prominently exhibit admixed western steppe ancestry [44], which spread over central Asia throughout the mid-second millennium BC [18–20].

## (b) Ancient DNA of livestock

We recovered *MT-CYB* sequences from 79 of 104 samples tested, indicating good preservation of ancient mtDNA (electronic supplementary material, table S2). Seventy-seven samples representing a cultural sequence from the Early Bronze Age to the medieval period yielded haplotypes that were identical to *MT-CYB* sequences specific to domesticated taxa, while two sequences were from *Capra sibirica* (electronic supplementary material, figure S2, dataset S1). We also found four conflicts between genetic species determinations and previously published [28] morphological identifications of Siberian ibex and domesticated sheep and goat. One domesticated sheep specimen each from Dali phase 1 and Begash phase 1a were directly $^{14}C$ dated, which tightly cluster with the other dates from their respective occupational strata (electronic supplementary material, tables S1 and S2). This demonstrates the earliest presence of domesticated sheep and goat in the central IAMC, since the first occupation of Dali (*ca* 2700 cal BC).

## (c) Isotopic reconstruction of steppe vegetation

Wild animal taxa from the study region exhibit dietary $\delta^{13}C$ values from between −26 and −22‰, demonstrating the predominance of $C_3$ floral taxa in the regional floral biome (electronic supplementary material, dataset S2) and contrasting starkly with $\delta^{13}C$ values of broomcorn millet, which

cluster at −12‰ (figure 2). Low $\delta^{13}C$ values from migratory saiga antelope and steppe tortoise indicate that drier environments, plausibly accessible to ancient livestock via transhumance to these areas, did not support detectable amounts of $C_4$ plants or regularly water-stressed $C_3$ plants enriched in $^{13}C$ [59,60]. These results indicate a $C_3$ environment in both the wetter mountains and drier open steppe, which reflect the present-day distribution of $C_3$ plants in the study region, consisting of mostly grasses (*Stipa* and *Festuca* spp.), followed by sedges (*Carex* spp.), forbs (*Artemesia* spp.) and shrubs (*Spiraea* spp. and *Caragana* spp.) [61].

## (d) Isotopic analysis of ancient pastoralist livestock

Intra-tooth isotopic sequences from Early Bronze Age domesticated sheep and cattle teeth exhibit high $\delta^{13}C_{apa}$ values from −7 to −3.2‰ that are notably coincident with low $\delta^{18}O_{apa}$ values associated with winter-season environmental inputs (figure 3a–c). Diverse seasonal $\delta^{13}C_{apa}$ values indicate that animal management practices were applied non-uniformly across herds (figure 3d–f). Overall, decreasing $\delta^{13}C_{apa}$ values leading into summer months after peak $C_4$ intake during winter strongly suggest a spring–summer dietary transition for sheep and goats to $C_3$ steppe plants present in montane and steppe vegetation communities. MixSIAR dietary models provide estimates of the seasonal relative importance of $C_4$ vegetation in livestock diets for the Early Bronze Age to Iron Age (figure 3g–i).

Sheep and goats from Dali demonstrate dietary intake with a substantial $C_4$ component by 2700 cal BC. Specifically, high $\delta^{13}C_{apa}$ values of *ca* −6‰ visible in some animals during winter months correspond to around 44–50% dietary intake of $C_4$ plants (figure 3a,d). Cattle from Dali exhibiting winter $\delta^{13}C_{apa}$ values of *ca* −7 to −8‰ correspond to about 5–15% $C_4$ dietary intake that changes little throughout the year, a pattern that persisted at the sites for millennia (figure 3f,i). Cattle require up to six times more feed by weight than caprines [15], so even a 10% $C_4$ diet represents a sizeable amount of $C_4$ biomass that must have been sourced from sufficiently large stockpiles regularly accessible to cattle.

The relative contribution of $C_4$ plants to sheep and goat diets subsequently increased during phase 1a of Begash (2345–2080 cal BC), indicated by winter $\delta^{13}C_{apa}$ values that reflect up to 68–74% $C_4$ diet (figure 3b,e). Over lifetime scales, ingestion of $C_4$ plants by livestock at Begash is also more pronounced than that observed at Dali during the third millennium BC (figure 3a,b). Over two-thirds of Begash livestock exhibit $\delta^{13}C_{col}$ values greater than −18‰, including a prominent subset of caprine values between *ca* −13 and −12‰ that reflect approximately 50–60% $C_4$ intake (figure 4b). As a point of reference, the domesticated sheep with the highest $\delta^{13}C_{apa}$ values in the dataset exhibited a $\delta^{13}C_{col}$ value of −16‰ that modelled to approximately 20% lifetime $C_4$ intake (figure 4b), demonstrating that animals with higher $\delta^{13}C_{col}$ values could have been provided with exclusively $C_4$ diets during winter.

For the earliest occupations of Begash, livestock consuming the most $C_4$ vegetation also exhibited high $\delta^{15}N_{col}$ values above 9‰ (figure 4b,c), indicated by significant positive correlations between $\delta^{13}C_{col}$ and $\delta^{15}N_{col}$ values for the Early and Middle Bronze Age phases (Pearson's $r = 0.6589$; $p < 0.0001$; $n = 34$ and Pearson's $r = 0.4610$; $p = 0.0006$; $n = 52$, respectively). In later phases at Begash, livestock with high $\delta^{15}N_{col}$

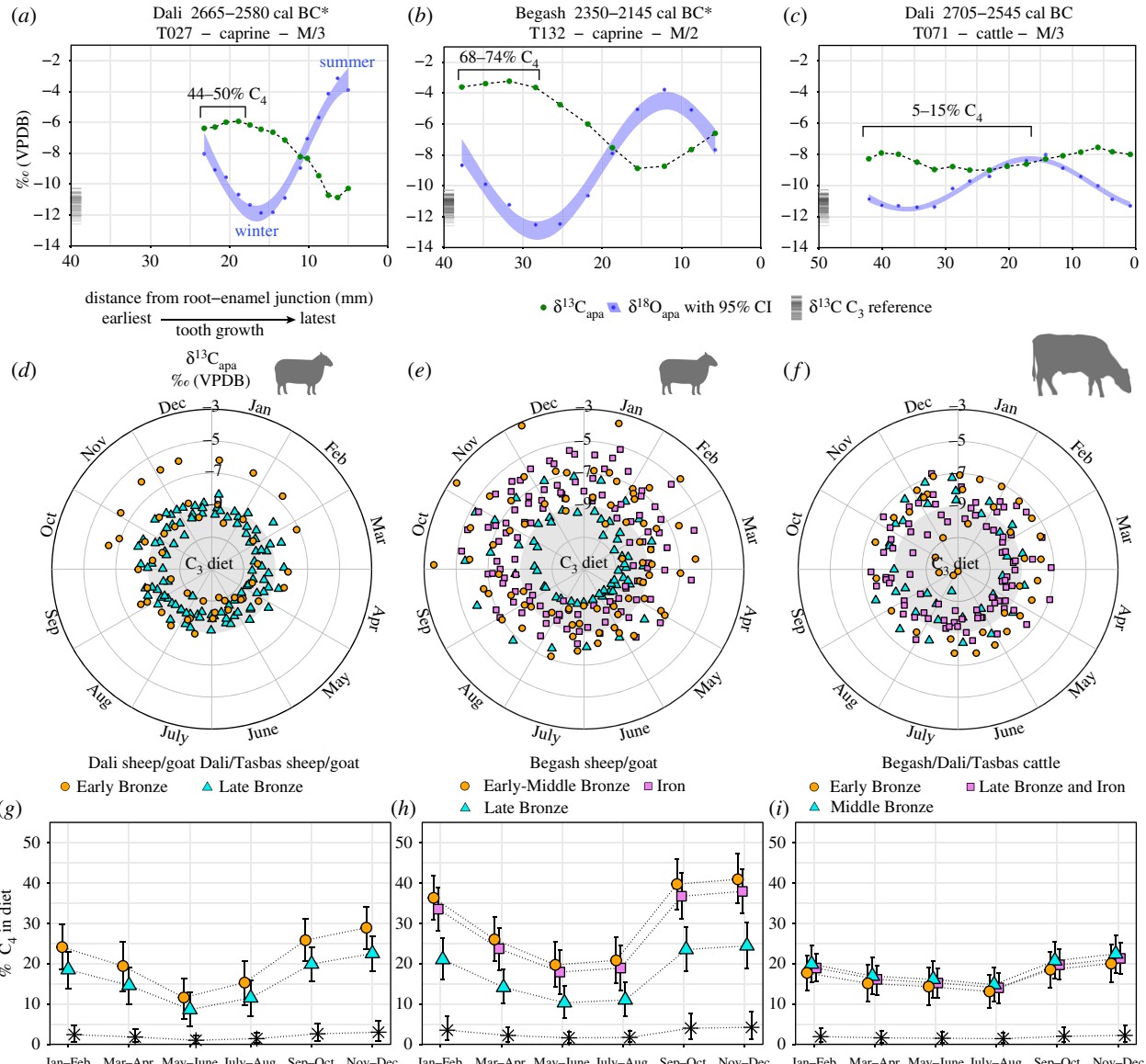

**Figure 3.** (*a–c*) Time series of $\delta^{13}C_{apa}$ and $\delta^{18}O_{apa}$ values from enamel bioapatite of second and third mandibular molars ($n = 42$). Symbol * denotes directly dated specimen. (*d–f*) $\delta^{13}C_{apa}$ values plotted as modelled annual cycles. (*g–i*) MixSIAR models showing relative proportions of $C_4$ plant consumption plotted in two-month periods with 90% credible intervals. Black asterisks represent herbivores with a $C_3$ diet. Individuals (electronic supplementary material, table S3; dataset S3): Dali caprines Early Bronze Age ($n = 4$); Dali and Tasbas caprines Late and Final Bronze Age ($n = 6$); Begash caprines Early-Middle Bronze Age ($n = 6$), Late Bronze Age ($n = 5$) and Iron Age ($n = 7$); Dali cattle Early Bronze Age ($n = 3$); Begash cattle Middle Bronze Age ($n = 3$). Dali, Begash, Tasbas cattle: Late and Final Bronze Age ($n = 8$). (Online version in colour.)

values above 9‰ tend to have low $\delta^{13}C_{col}$ values *ca* −19‰ (electronic supplementary material, figure S5). From all sampled phases at Begash, most livestock exhibit higher $\delta^{15}N_{col}$ values than that of wild herbivores by approximately 4‰ (Mann–Whitney $U = 2690$; $p < 0.0001$; $n_{livestock} = 153$; $n_{wild} = 21$). This demonstrates not only long-term modifications of local ecosystems, due to pasturing and farming practices involving manure deposition, thus enriching vegetation in $^{15}N$, but also shows shared ecological niches between wild herbivores and livestock both exhibiting low $\delta^{13}C_{col}$ and $\delta^{15}N_{col}$ values.

A conspicuous decline in $C_4$ dietary intake for caprines took place from the middle to late second millennium BC (figures 3*g*,*h* and 4*c*,*d*; electronic supplementary material, S5). By this time, however, direct human dietary intake of millet is common in the IAMC and adjacent regions, evidenced by a conspicuous rise in human $\delta^{13}C_{col}$ values first visible at approximately 2100 cal BC in an individual from

Kanai in the Kazakh Altai with a $\delta^{13}C_{col}$ value of −14.5‰ (electronic supplementary material, figures S6 and S7). Unfortunately, human samples from the third millennium BC are too few to know whether early $C_4$ plant intake by livestock at Dali or Begash coincided with direct human millet consumption (electronic supplementary material, figures S6 and S7). Strikingly, winter levels of livestock $C_4$ dietary intake during the Iron Age at Begash are equally elevated compared to that of the site's Early and Middle Bronze Age occupations (figure 3*h*).

## 5. Discussion

### (a) Millet foddering of Early Bronze Age livestock
While pastoralist livestock are known to alter the relative abundance of plant taxa in grasslands [62] and local nitrogen cycles [42], we are confident that wild $C_4$ plants are not

Proc. R. Soc. B 286: 20191273

**6**

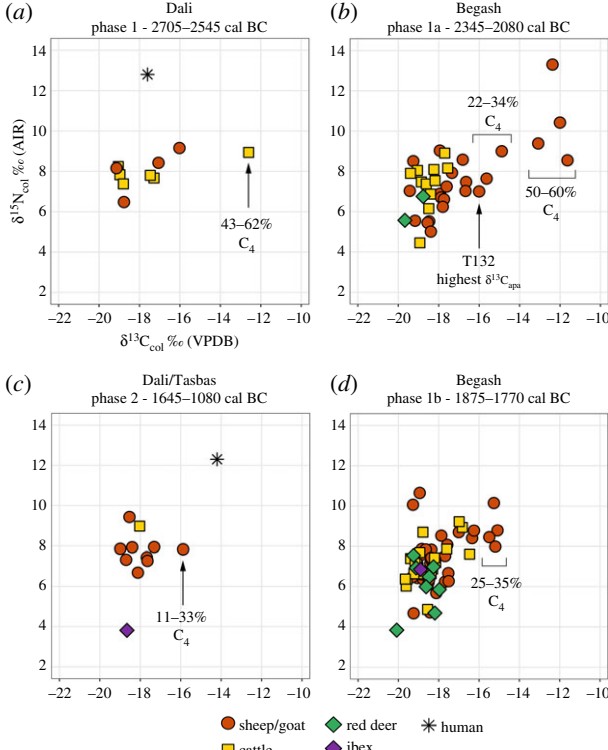

**Figure 4.** (a–d) $\delta^{13}C_{col}$ and $\delta^{15}N_{col}$ values of livestock, wild herbivores and humans from sampled sites. Isotopic data from Late Bronze Age and Iron Age periods are displayed in electronic supplementary material, figure S4. See electronic supplementary material, table S4 for sample summary and yields and dataset S4 for raw data. (Online version in colour.)

contributing to high $\delta^{13}C$ values in livestock based on (1) site locations in the mountains where wild $C_4$ plants do not grow [60] and (2) consistently low $\delta^{13}C$ values in wild herbivorous taxa with dietary niches overlapping the more arid lowlands where wild $C_4$ plants might be expected. Thus, relatively high $\delta^{13}C$ values in ancient livestock are the direct outcome of pastoralists foddering animals with cultivated millet, which would have greatly enhanced the efficiency and resiliency of food production in the harsh winters of Inner Asia.

One unlikely scenario to explain anticorrelation of $\delta^{13}C_{apa}$ and $\delta^{18}O_{apa}$ values in caprines would be herding over 1000 km to the west of the Dzhungar Mountains to the Kyzylkum Desert (figure 1), where wild $C_4$ plants are present in low densities [63]. However, our finding of steady and low levels of seasonal $C_4$ dietary intake (approx. 10–20%) in cattle suggests that pastoralists primarily managed cattle near millet fodder reserves at settlements, where abundant mountain run-off provides ample water for these obligate-drinking livestock, especially if exploited for milk. A far more likely scenario is that caprines, compared to cattle, received more millet fodder in winter as a dietary percentage and in summer were herded over a wider local range away from millet resources to lowlands or to alpine pastures, causing large seasonal shifts in $\delta^{13}C_{apa}$ values for caprines (figure 3g,h).

Similar intensities of millet winter foddering between Begash's Early Bronze and Iron Age occupations suggests that third millennium BC agro-pastoral strategies reached high levels of local production, despite different social conditions. The Iron Age throughout the foothill zone of southeastern Kazakhstan is well characterized by abundant seed remains of broomcorn millet [3], which benefits from

this region's adequate summer heat for productive cultivation [25,64]. Mixed herding and multi-crop farming in the IAMC during the Iron Age are associated with diverse communities commonly classified as Scythian/Saka and Wusun agro-pastoralists [6,12,65], exhibiting amplified social hierarchies and protracted cultural and political networks that arguably involved excess agricultural output [3,15]. For Early Bronze Age pastoralists in the IAMC, however, millet would have been initially perceived as an attractive fodder resource offering hardy and versatile properties, especially compared to wheat and barley, which might have already been locally cultivated. The subsequent decline in millet foddering during the Late Bronze Age is likely due to $C_3$ cultigens arriving via secondary crop dispersals [5], especially landraces of barley adapted to high altitudes [66]. From *ca* 1750–1000 cal BC, this diversification in cropping across Asia is also linked to reducing risks during climatic shifts towards cooler conditions [25]. However, improved palaeo-climate records are needed to characterize earlier periods.

Our finding that not all livestock individuals exhibited high $\delta^{13}C$ values corresponds with ethnographic accounts of steppe pastoralists exercising adaptable strategies of food production combining mixed herding and plant cultivation to balance graze and fodder availability with goals for dairy and fibre production, meat and fat harvesting, herd security and wealth generation [15,67–69]. Versatility in animal management practices is especially pronounced when considering the variation in $\delta^{15}N_{col}$ values of caprines from the first phase of Begash (figure 4b), which also points towards the recycling of nutrients in millet cultivation plots using manure. However, contemporaneous and later livestock with low $\delta^{13}C_{col}$ values also exhibit a wide ranging $\delta^{15}N_{col}$ values (figure 4b,d), consistent with variable growing conditions for $C_3$ crops and/or irregular stocking rates on $C_3$ steppe pastures characterized by uneven nitrogen pools [42,70].

Given the case for millet foddering at Begash during its earliest occupation, millet cultivation at Dali by *ca* 2700 cal BC emerges as a credible scenario. High $\delta^{13}C_{apa}$ values reflecting winter dietary intake of livestock from phase 1 Dali strongly contrast against the carbon isotopic composition of surrounding $C_3$ steppe vegetation. That early millet foddering at Dali was followed by more prominent millet foddering of caprines at Begash several hundred years later suggests a rapid process of integrating pastoralism and millet agriculture, while herding strategies sustained a component of grazing livestock on $C_3$ steppe vegetation. Taken together, our data illustrate that communities in the IAMC since the Early Bronze Age were engaged in ecologically dynamic subsistence strategies using mixed livestock herding and intermittent millet cultivation, likely involving flexible herd mobility patterns of seasonal transhumant moves and prolonged settlement residence.

## (b) Domesticated bovids enhanced millet transmission

Early genetic evidence for domesticated sheep and goat and a human individual with admixed northeastern Iranian and Eurasian hunter–gatherer ancestries at Dali allow us to consider the transmission of pastoralism to Inner Asia as the combined effects of mobility, communication and exchange with agro-pastoralists in the southern reaches of the IAMC [23,44,71]. However, we cannot rule out simultaneous or earlier transmissions of livestock to the Dzhungar and Tian Shan

mountains from Afanasievo or other Eneolithic communities in the Altai. Nevertheless, the integration of millet cultivation with livestock management at Dali and Begash implies that Early Bronze Age pastoralists were already well connected throughout the foothills of western China, perhaps as far east as Gansu, and accelerated the transmission of millet westward. The addition of millet agriculture to pastoralism likely facilitated new labour divisions and expanded social networks, reflecting an important medium for the inter-regional transfer of food technologies.

Prior to the arrival of pastoralist subsistence, millet may have been dispersed slowly across northern China by settled agriculturalists, who were without indigenously domesticated herbivores but instead had domesticated pigs, which are managed close to settlements and do not afford easy mobility as do sheep, goat and cattle. It would be unsurprising if future research in western China recovered millet remains associated with pastoralist livestock at least several centuries earlier than 2700 cal BC. Current human palaeogenetic studies in Eurasia invoke the so-called massive migrations of archaeologically defined populations to explain changes in the genetic composition of ancient peoples and shifts in their food production [18–21]. Our finding of swift millet transmission at Dali and Begash prior to influxes of people with western steppe ancestry (Yamnaya or Afanasievo) emphasizes a subsistence foundation for extensive social connectivity as part of more localized interactions and mobility, underscoring the substantial time depth and multi-directionality of cultural transmissions via pastoralist societies.

## 6. Conclusion

Combined genetic and stable isotopic analyses of faunal skeletal remains from settlement sites in the IAMC reveal the earliest evidence in Inner Asia (ca 2700 cal BC) for domesticated sheep, goat, and cattle and their intensive management by way of foddering with cultivated millet. In the context of contemporaneous mortuary rituals using domesticated plants [11–13], our findings suggest that Early Bronze Age pastoralists in the IAMC were immortalizing the economic significance of plant cultivation for enhancing livestock production, as they also contributed to the spread of millet and knowledge about its production westward. Early and precocious use of millet to fodder livestock by IAMC pastoralists suggests there may be a fundamentally fast pace to transmissions of domesticated plants by people exploiting sheep, goat and cattle. Our methodological approach focused on faunal remains provides a critical means to test this model for millets in other regions of Eurasia and for other $C_4$ crops, such as sorghum in Africa, at both local and continental scales.

Data accessibility. Genetic data are available in electronic supplementary material, dataset S1. Isotopic data are available in electronic supplementary material, datasets S2–S5.
Authors' contributions. T.R.H., M.D.F., P.N.D.D., and A.M. designed and performed archaeological research at Dali. T.R.H. designed and performed genetic and isotopic research. T.R.H. and C.A.M. interpreted isotopic data. T.R.H. and A.N. interpreted genetic data. T.R.H. wrote the manuscript with guidance from C.A.M. and M.D.F. All authors contributed critically to subsequent drafts and gave final approval for publication.
Competing interests. We declare we have no competing interests.
Funding. This research was supported by the doctoral fellowship of T.R.H. at the Graduate School 'Human Development in Landscapes' (Supervisors: C.A.M. and A.N.; German Research Foundation: GSC 208). Excavations of Dali were funded by Washington University in St Louis (M.D.F.) and the U.S. National Science Foundation (no. 1132090; P.N.D.D. and M.D.F.). This research is part of a project that has received funding from the European Research Council (ERC) under the European Union's Horizon 2020 research and innovation programme (Grant agreement no. 772957; 'ASIAPAST' held by C.A.M.).
Acknowledgements. We greatly appreciate the support of B. Baitanayev, director of the Margulan Institute of Archaeology, and B. Kakabaev, former deputy director of the Central State Museum of the Republic of Kazakhstan; archaeological specimens were sampled from these institutions under permit nos. 54/20-202 and 1-10-224, respectively. We thank D. Voyakin for access to ancient saiga specimens. We also thank N. Andersen (Leibniz Labor, Kiel University) for assistance with carbon and oxygen mass spectrometry of bioapatite carbonates and R. Michener (Boston University Stable Isotope Laboratory) for assistance with carbon and nitrogen mass spectrometry of bone collagen. We thank the three anonymous reviewers for constructive criticism.

## Endnote

[1]Dzhungar Mountain Archaeology Project, Co-PIs: M.D.F. and A.M.

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
