## [Reviewer comments · Proceedings of the Royal Society B: Biological Sciences]

Review History

RSPB-2019-1273.R0 (Original submission)

Review form: Reviewer 1 (Giedre Motuzaitė Matuzevičiūtė)

Recommendation

Accept with minor revision (please list in comments)

Scientific importance: Is the manuscript an original and important contribution to its field?

Excellent

General interest: Is the paper of sufficient general interest?

Excellent

Quality of the paper: Is the overall quality of the paper suitable?

Excellent

Is the length of the paper justified?

Yes

Should the paper be seen by a specialist statistical reviewer?

No

Do you have any concerns about statistical analyses in this paper? If so, please specify them explicitly in your report.

No

It is a condition of publication that authors make their supporting data, code and materials available - either as supplementary material or hosted in an external repository. Please rate, if applicable, the supporting data on the following criteria.

Is it accessible?

No

Is it clear?

N/A

Is it adequate?

N/A

Do you have any ethical concerns with this paper?

No

Comments to the Author

The RSPB-2019-1273 manuscript is extremely important contribution to our current knowledge, allowing to understand not only when but also how the earliest food globalisation processes took place across Eurasia. This discovery pinpoints the earliest contact routes that initially linked the western and eastern Eurasia, paving the roads of interaction for future historical networks such as Silk Road. This manuscript is of outstanding importance, scientifically solid and it will be heavily cited in the future, therefore I strongly recommend to a journal accepting it for publication.

I have a few recommendations for the authors to consider while preparing the final submission of this work.

Firstly, I would advise them to sharpen the abstract, pointing out more clearly the significance of this work and the application of novel methodology for the first time to pinpoint the earliest consumption of cultivated crops by domestic cattle, at the same time emphasizing the importance of understanding these processes in central Asia. In addition, in the same abstract the sentence on ritual use of crops should be removed/rephrased as previous studies had only suggested it as a possibility, but no one has actively studied it. In addition, the authors do not really talk about it later on in the text and do not contrast their discoveries against the ritual use of plants. It is still puzzling to me how come no crop chaff remains were found in the early sites of Dzhungaria if potentially ovicaprids were eating it. If there is any phytolith work please mention. Can you elaborate on it instead of calling it "exotica"?

In the text:

Hordeum vulgare normally means Hulled barley, while the earliest barley in the region is naked varieties *Hordeum vulgare* var. *nudum*. So I would suggest just using *Hordeum* spp. or just barley to be correct. In addition, I would also stay away from contracting nomadic central Asian populations with other societies living on the other ends of Eurasia as if the central Asian ones are somehow unique. Firstly, no study has really provided evidence on how mobile they really were, whether all population were mobile or just a few shepherds were moving around, and the rest are staying in winter camps all year round and growing crops. Also, there are semi mobile

populations pretty much everywhere where there are domestic ovicaprids and cattle (Near East including), so central Asia in this sense is not unique.

Also to be precise in the scientific publication, I would recommend to avoid using words “identical” (line 109), maybe better statistically the same; “massive migration” (line 401), how massive is massive?

Unfortunately, I was not able to see the supplementary material, but hope somewhere authors explain how they calculate the percentages of millet dietary intake, this is important to explain to the reader.

Few more comments:

Reference “22” is not the best here

I do not understand sentence in Line 245-247. Also please use δ next to 13C

Various water stress experiments on plants also on plants growing in higher elevations do increase the δ 13C values. Maybe authors have to mention this in the text in a few sentences.

Is it not better to use wild boar instead of wild pig in Fig. 2? Also it is not clear from the graph or the described methodology how did you make enamel compatible to collagen. Please add the sentence in methodology on it (unless it is in SOM).

In the text, in methodology is not clear if you are making the calculations on C4 dietary intake from collagen or apatite, and if from both, how do you calculate the offset values.

There is just a minor comments and the rest of methodology is clear and solid.

In the discussion, if you are talking about early iron age populations, the Scythians should not be the only people group that deserve mentioning. I also would ask you to clarify sentence in lines 391-392.

Maybe the authors in the discussions could also elaborate how millet got all the way to Kazakhstan territory without the aid of pastoral societies (as ancient Chinese did not domesticate cattle or sheep/ goat) and that pastoralism is not the only mean of crop translocation.

The separate conclusion will allow to pinpoint the outstanding contribution of this article.

Review form: Reviewer 2 (Sandra L. Olsen)

Recommendation

Accept as is

Scientific importance: Is the manuscript an original and important contribution to its field?

Excellent

General interest: Is the paper of sufficient general interest?

Excellent

Quality of the paper: Is the overall quality of the paper suitable?

Excellent

Is the length of the paper justified?

Yes

Should the paper be seen by a specialist statistical reviewer?

Yes

Do you have any concerns about statistical analyses in this paper? If so, please specify them explicitly in your report.

No

It is a condition of publication that authors make their supporting data, code and materials available - either as supplementary material or hosted in an external repository. Please rate, if applicable, the supporting data on the following criteria.

Is it accessible?

Yes

Is it clear?

Yes

Is it adequate?

Yes

Do you have any ethical concerns with this paper?

No

Comments to the Author

I find this manuscript acceptable for publication just as it is.

Review form: Reviewer 3

Recommendation

Accept with minor revision (please list in comments)

Scientific importance: Is the manuscript an original and important contribution to its field?

Excellent

General interest: Is the paper of sufficient general interest?

Good

Quality of the paper: Is the overall quality of the paper suitable?

Good

Is the length of the paper justified?

Yes

Should the paper be seen by a specialist statistical reviewer?

No

Do you have any concerns about statistical analyses in this paper? If so, please specify them explicitly in your report.

No

It is a condition of publication that authors make their supporting data, code and materials available - either as supplementary material or hosted in an external repository. Please rate, if applicable, the supporting data on the following criteria.

Is it accessible?

Yes

Is it clear?

Yes

Is it adequate?

Yes

Do you have any ethical concerns with this paper?

No

Comments to the Author

Using stable isotope and aDNA data, this article argues that initial domesticated crops, specifically millet, were cultivated and dispersed by pastoralists along the Inter Asian Mountain Corridor. This paper is of high quality and scientific importance. The methodology is solid and the overall argument is persuasive. Most of my suggestions are for how the authors could make the paper more accessible to readers who do not work in this region. The archaeological background and discussion and conclusion sections are especially in need of more contextual information to make this article of general interest to larger audiences.

1. Introduction. The opening paragraph is a little misleading in that it emphasizes the use of initial crop domesticates in mortuary contexts over the argument that is explored throughout the rest of the paper- that pastoralists spread these crops via their agricultural/economic processes. The authors might consider moving this part (lines 66-71) to the background section as part of their larger discussion of prehistoric development in the IAMC.
2. Archaeological Background. If the argument is that pastoralism precedes millet cultivation, it would make more sense to present the material chronologically and reverse the two paragraphs in this section (so paragraph 2 before paragraph 1). This rearrangement would help to establish the archaeological context of the region before introducing the specific goals of this paper. The authors might also consider similarly rearranging the introduction to help with flow and clarity (and see above).
3. Materials and Methods.
 - a. line 126: Could you briefly elaborate on how previous faunal data clearly indicate intensive exploitation of pastoralist livestock?
 - b. Considering the wide range of material that was analyzed for stable isotopes (collagen, apatite, enamel, hair), I would appreciate a brief explanation either in the paper or supplemental material of how or why they can be directly compared. I'm specifically thinking of Figure 2 where all of these materials are graphed using the same y-axis.
4. Results (stable isotope analysis).
 - a. The cattle result are very interesting too. Did you expect them to have different isotopic signatures and/or include them as a control for detecting transhumance? Were they as important to the economy as sheep/goat?
 - b. lines 273-277: What point are you trying to make in this sentence?
 - c. line 297-298: What are some examples of these "long-term agro-pastoral modifications of local ecosystems"?

d. Why does livestock consumption of millet decline as human consumption increases in mid-late 2nd millennium? And then increase again in Iron Age Begash?

5. Discussion and Conclusion.

- a. lines 344-348: Again, why would cattle be treated differently?
- b. lines 366-372: Are there other possible (non-anthropogenic) explanations for the variable nitrogen values? What is the climate like during this time period?
- c. line 380ish: Are you arguing that goats/sheep at Dali were not as transhumant as elsewhere?
- d. lines 383-385: What exactly does this mean? The potential links between these agropastoral practices and processes of cultural transmission should be better explained. In general this paper does a great job of interpreting results at the site/subsistence level, but doesn't fully explain the larger significance of these results or how they might articulate with other developments in this region/time period.

6. Small Details

- a. line 53: Sentence is missing "of"
- b. lines 95-99: This sentence should be rewritten for clarity
- c. line 312: Figure 4H?
- d. lines 333-334: "alter" used twice in one sentence
- e. line 362: "Corresponds" or something similar instead of "resonate"
- f. Both BC and BCE are used at different points
- g. The use of "agronomic" as a descriptor for economic practices throughout this paper is not one I'm familiar with (American English). A different word would be better.
- h. Supplemental Files 4-6 should be standardized so that %C, %N, and C:N results are included in all of them.

Decision letter (RSPB-2019-1273.R0)

01-Jul-2019

Dear Dr Hermes:

Your manuscript has now been peer reviewed and the reviews have been assessed by an Associate Editor. The reviewers' comments (not including confidential comments to the Editor) and the comments from the Associate Editor are included at the end of this email for your reference. As you will see, the reviewers and the Editors have raised some concerns with your manuscript and we would like to invite you to revise your manuscript to address them.

When submitting your revision please upload a file under "Response to Referees" - in the "File Upload" section. This should document, point by point, how you have responded to the

reviewers' and Editors' comments, and the adjustments you have made to the manuscript. We require a copy of the manuscript with revisions made since the previous version marked as 'tracked changes' to be included in the 'response to referees' document.

Research ethics:

Use of animals and field studies:

If you wish to submit your data to Dryad (<http://datadryad.org/>) and have not already done so you can submit your data via this link [http://datadryad.org/submit?journalID=RSPB&manu=\(Document not available\)](http://datadryad.org/submit?journalID=RSPB&manu=(Document%20not%20available)), which will take you to your unique entry in the Dryad repository.

Online supplementary material will also carry the title and description provided during

submission, so please ensure these are accurate and informative. Note that the Royal Society will not edit or typeset supplementary material and it will be hosted as provided. Please ensure that the supplementary material includes the paper details (authors, title, journal name, article DOI). Your article DOI will be 10.1098/rspb.[paper ID in form xxxx.xxxx e.g. 10.1098/rspb.2016.0049].

Please submit a copy of your revised paper within three weeks. If we do not hear from you within this time your manuscript will be rejected. If you are unable to meet this deadline please let us know as soon as possible, as we may be able to grant a short extension.

Best wishes,

The Proceedings B Team
mailto:proceedingsb@royalsociety.org

Comments to Author:

Your study has been seen by 3 expert reviewers and all agree that there is interesting and sound science here that is appropriate for Proc B's broad audience, which I concur with. There are numerous very constructive points to account for in revisions, but all of these seem very achievable. We look forward to seeing your revised manuscript.

Kind regards

Prof. John Hutchinson, Editor

Reviewer(s)' Comments to Author:

Referee: 1

Comments to the Author(s)

The RSPB-2019-1273 manuscript is extremely important contribution to our current knowledge, allowing to understand not only when but also how the earliest food globalisation processes took place across Eurasia. This discovery pinpoints the earliest contact routes that initially linked the western and eastern Eurasia, paving the roads of interaction for future historical networks such as Silk Road. This manuscript is of outstanding importance, scientifically solid and it will be heavily cited in the future, therefore I strongly recommend to a journal accepting it for publication.

I have a few recommendations for the authors to consider while preparing the final submission of this work.

Firstly, I would advise them to sharpen the abstract, pointing out more clearly the significance of this work and the application of novel methodology for the first time to pinpoint the earliest consumption of cultivated crops by domestic cattle, at the same time emphasizing the importance of understanding these processes in central Asia. In addition, in the same abstract the sentence on ritual use of crops should be removed/rephrased as previous studies had only suggested it as a possibility, but no one has actively studied it. In addition, the authors do not really talk about it later on in the text and do not contrast their discoveries against the ritual use of plants. It is still puzzling to me how come no crop chaff remains were found in the early sites of Dzhungaria if potentially ovicaprids were eating it. If there is any phytolith work please mention. Can you elaborate on it instead of calling it "exotica"?

In the text:

Hordeum vulgare normally means Hulled barley, while the earliest barley in the region is naked varieties *Hordeum vulgare* var. *nudum*. So I would suggest just using *Hordeum* spp. or just barley to be correct. In addition, I would also stay away from contracting nomadic central Asian populations with other societies living on the other ends of Eurasia as if the central Asian ones are somehow unique. Firstly, no study has really provided evidence on how mobile they really were, whether all population were mobile or just a few shepherds were moving around, and the rest are staying in winter cam all year round and growing crops. Also, there are semi mobile populations pretty much everywhere where there are domestic ovicaprids and cattle (Near East including), so central Asia in this sense is not unique.

Also to be precise in the scientific publication, I would recommend to avoid using words "identical" (line 109), maybe better statistically the same; "massive migration" (line 401), how massive is massive?

Unfortunately, I was not able to see the supplementary material, but hope somewhere authors explain how they calculate the percentages of millet dietary intake, this is important to explain to the reader.

Few more comments:

Reference "22" is not the best here

I do not understand sentence in Line 245-247. Also please use δ next to 13C

Various water stress experiments on plants also on plants growing in higher elevations do increase the δ 13C values. Maybe authors have to mention this in the text in a few sentences.

Is it not better to use wild boar instead of wild pig in Fig. 2? Also it is not clear from the graph or the described methodology how did you make enamel compatible to collagen. Please add the sentence in methodology on it (unless it is in SOM).

In the text, in methodology is not clear if you are making the calculations on C4 dietary intake from collagen or apatite, and if from both, how do you calculate the offset values.

There is just a minor comments and the rest of methodology is clear and solid.

In the discussion, if you are talking about early iron age populations, the Scythians should not be the only people group that deserve mentioning. I also would ask you to clarify sentence in lines 391-392.

Maybe the authors in the discussions could also elaborate how millet got all the way to Kazakhstan territory without the aid of pastoral societies (as ancient Chinese did not domesticate cattle or sheep/goat) and that pastoralism is not the only mean of crop translocation.

The separate conclusion will allow to pinpoint the outstanding contribution of this article.

Referee: 2

Comments to the Author(s)

I find this manuscript acceptable for publication just as it is.

Referee: 3

Comments to the Author(s)

Using stable isotope and aDNA data, this article argues that initial domesticated crops, specifically millet, were cultivated and dispersed by pastoralists along the Inter Asian Mountain

Corridor. This paper is of high quality and scientific importance. The methodology is solid and the overall argument is persuasive. Most of my suggestions are for how the authors could make the paper more accessible to readers who do not work in this region. The archaeological background and discussion and conclusion sections are especially in need of more contextual information to make this article of general interest to larger audiences.

1. Introduction. The opening paragraph is a little misleading in that it emphasizes the use of initial crop domesticates in mortuary contexts over the argument that is explored throughout the rest of the paper- that pastoralists spread these crops via their agricultural/economic processes. The authors might consider moving this part (lines 66-71) to the background section as part of their larger discussion of prehistoric development in the IAMC.

2. Archaeological Background. If the argument is that pastoralism precedes millet cultivation, it would make more sense to present the material chronologically and reverse the two paragraphs in this section (so paragraph 2 before paragraph 1). This rearrangement would help to establish the archaeological context of the region before introducing the specific goals of this paper. The authors might also consider similarly rearranging the introduction to help with flow and clarity (and see above).

3. Materials and Methods.

a. line 126: Could you briefly elaborate on how previous faunal data clearly indicate intensive exploitation of pastoralist livestock?

b. Considering the wide range of material that was analyzed for stable isotopes (collagen, apatite, enamel, hair), I would appreciate a brief explanation either in the paper or supplemental material of how or why they can be directly compared. I'm specifically thinking of Figure 2 where all of these materials are graphed using the same y-axis.

4. Results (stable isotope analysis).

a. The cattle result are very interesting too. Did you expect them to have different isotopic signatures and/or include them as a control for detecting transhumance? Were they as important to the economy as sheep/goat?

b. lines 273-277: What point are you trying to make in this sentence?

c. line 297-298: What are some examples of these "long-term agro-pastoral modifications of local ecosystems"?

d. Why does livestock consumption of millet decline as human consumption increases in mid-late 2nd millennium? And then increase again in Iron Age Begash?

5. Discussion and Conclusion.

a. lines 344-348: Again, why would cattle be treated differently?

b. lines 366-372: Are there other possible (non-anthropogenic) explanations for the variable nitrogen values? What is the climate like during this time period?

c. line 380ish: Are you arguing that goats/sheep at Dali were not as transhumant as elsewhere?

d. lines 383-385: What exactly does this mean? The potential links between these agropastoral practices and processes of cultural transmission should be better explained. In general this paper does a great job of interpreting results at the site/subsistence level, but doesn't fully explain the larger significance of these results or how they might articulate with other developments in this region/time period.

6. Small Details

a. line 53: Sentence is missing "of"

b. lines 95-99: This sentence should be rewritten for clarity

c. line 312: Figure 4H?

d. lines 333-334: "alter" used twice in one sentence

- e. line 362: "Corresponds" or something similar instead of "resonate"
- f. Both BC and BCE are used at different points
- g. The use of "agronomic" as a descriptor for economic practices throughout this paper is not one I'm familiar with (American English). A different word would be better.
- h. Supplemental Files 4-6 should be standardized so that %C, %N, and C:N results are included in all of them.

Author's Response to Decision Letter for (RSPB-2019-1273.R0)

See Appendix A.

RSPB-2019-1273.R1 (Revision)

Review form: Reviewer 1 (Giedre Motuzaitė Matuzeviciute)

Recommendation

Accept as is

Scientific importance: Is the manuscript an original and important contribution to its field?

Excellent

General interest: Is the paper of sufficient general interest?

Excellent

Quality of the paper: Is the overall quality of the paper suitable?

Excellent

Is the length of the paper justified?

Yes

Should the paper be seen by a specialist statistical reviewer?

No

Do you have any concerns about statistical analyses in this paper? If so, please specify them explicitly in your report.

No

It is a condition of publication that authors make their supporting data, code and materials available - either as supplementary material or hosted in an external repository. Please rate, if applicable, the supporting data on the following criteria.

Is it accessible?

Yes

Is it clear?

Yes

Is it adequate?

Yes

Do you have any ethical concerns with this paper?

No

Comments to the Author

Publish as it is, the authors have done great job responding to our comments and have improved the manuscript significantly.

Review form: Reviewer 2

Recommendation

Accept as is

Scientific importance: Is the manuscript an original and important contribution to its field?

Excellent

General interest: Is the paper of sufficient general interest?

Excellent

Quality of the paper: Is the overall quality of the paper suitable?

Excellent

Is the length of the paper justified?

Yes

Should the paper be seen by a specialist statistical reviewer?

Yes

Do you have any concerns about statistical analyses in this paper? If so, please specify them explicitly in your report.

No

It is a condition of publication that authors make their supporting data, code and materials available - either as supplementary material or hosted in an external repository. Please rate, if applicable, the supporting data on the following criteria.

Is it accessible?

Yes

Is it clear?

Yes

Is it adequate?

Yes

Do you have any ethical concerns with this paper?

No

Comments to the Author

I found this research to be quite timely and relevant. It is thorough and integrates techniques in such a way that considerable valuable information can be gleaned. This is excellent work!

Decision letter (RSPB-2019-1273.R1)

05-Aug-2019

Dear Dr Hermes

I am pleased to inform you that your manuscript entitled "Early integration of pastoralism and millet cultivation in Bronze Age Eurasia" has been accepted for publication in Proceedings B. Congratulations!! The reviewers and editorial board were truly impressed.

Open Access

You are invited to opt for Open Access, making your freely available to all as soon as it is ready for publication under a CCBY licence. Our article processing charge for Open Access is £1700. Corresponding authors from member institutions (<http://royalsocietypublishing.org/site/librarians/allmembers.xhtml>) receive a 25% discount to these charges. For more information please visit <http://royalsocietypublishing.org/open-access>.

Paper charges

Sincerely,

Professor John Hutchinson
Editor, Proceedings B
mailto: proceedingsb@royalsociety.org

Appendix A

RSPB-2019-1273

Early integration of pastoralism and millet cultivation in Bronze Age Eurasia
Hermes et al.

--

Dear Reviewers,

Thank you for your thoughtful and helpful comments for improving our manuscript. Below, we copied your points of criticism and highlighted this text in grey. After each point, we responded in normal text. The “tracked changes” versions of the manuscript and SI are at the end of this response document.

Both Reviewers 1 and 3 commented on our calculation of dietary $\delta^{13}\text{C}$ values from $\delta^{13}\text{C}$ values of different tissues from different taxa, allowing these data to be displayed on a single Y-axis in figure 2. We have the diet-tissue spacing factors for making these conversions in the Supplementary Information and relevant citations that establish their validity. Since this information is somewhat tedious to read in the manuscript and the journal has strict page limits, which we are at risk of exceeding, we decided to place this information in the SI. To clarify, we emphasized in the manuscript’s Methods section and in the caption of figure 2 that the diet-tissue spacing factors are in SI text 2. We hope this balances the desire to have this information accessible with the consideration for not taking up too much space in the manuscript.

I would like to point out that I found some minor counting errors of our raw data that touched on the presentation of aDNA sequence recovery and statistical analysis of the isotopic data. Specifically, one aDNA sequence was counted twice, bringing the revised total number of recovered cytochrome *b* sequences to 79 from 80, which also changed the number of recovered cytochrome *b* sequences of domesticated sheep and goat to 77 from 78. Furthermore, the taxonomic classification of one Siberian ibex specimen for the isotopic analysis was incorrectly marked as a domesticated caprine, which affected figure 4*b* and the calculation of Pearson’s *r* and Mann-Whitney *U*. The numerical differences were negligible, for example, a p-value shifting by 0.0001. I scrutinized the entire dataset of our study for other errors and reran the analyses for inconsistencies. I did not find further issues.

I decided to prune out a subset of human isotopic data displayed in figure S6-7 (now, dataset S5) because these were predominately dated to the later Iron Age and medieval period and were inconsistent with the chronological scope of our study. The change reduces the number of human samples from 220 to 174. This reduction does not affect the main finding of this analysis, which is that human isotopic data are too sparse to evaluate the uptake of millet as a dietary component by people in the steppe zone during the third millennium BC. The data reduction also removed some references, helping our paper fall below the 10-page limit.

We hope our responses below and improvements to the manuscript satisfy your concerns with our study. Thank you very much for your time and effort reviewing our paper.

Sincerely,

Taylor Hermes

--

Referee: 1

Firstly, I would advise them to sharpen the abstract, pointing out more clearly the significance of this work and the application of novel methodology for the first time to pinpoint the earliest consumption of cultivated crops by domestic cattle, at the same time emphasizing the importance of understanding these processes in central Asia. In addition, in the same abstract the sentence on ritual use of crops should be removed/rephrased as previous studies had only suggested it as a possibility, but no one has actively studied it. In addition, the authors do not really talk about it later on in the text and do not contrast their discoveries against the ritual use of plants. It is still puzzling to me how come no crop chaff remains were found in the early sites of Dzhungaria if potentially ovicaprids were eating it. If there is any phytolith work please mention. Can you elaborate on it instead of calling it "exotica"?

Reply: Thank you for the suggestion to improve the visibility of the significance of our study in the abstract. Here, we made edits to downplay the ritual use of the seeds by putting this information into a separate sentence where we indicate to the reader that this is simply the best we know about how these crops were used, while also explaining that the mortuary context of the seeds does not inform on cultivation or subsistence. We also added a clause to the end of the abstract explaining that our study suggests that pastoralist livestock were critical for the westward transmission of millet in order to introduce the broader significance of the work. In the Discussion, this argument is fleshed out in the context of early Chinese societies developing millet agriculture without domesticated herbivores, so it is the arrival of these livestock that facilitates the spread of millet agriculture.

To date, there is little phytolith work done on sites in the Dzhungar Mountains. A small study was conducted by Breadmore and published in Doumani et al. 2015. The results of the phytolith work do not inform on the use of domesticated plants due to significant overlap of phytolith morphotypes between plant families that include cultivated crops and wild taxa. Thus, it is not worthwhile to mention this work and would require a considerable amount of text to explain why. Overall, the absence of crop chaff is indeed intriguing. Presumably, these plant parts could have been deposited archaeologically through the burning of dung from animals foddered with crops (if they were fed stalks with seeds), however, floatation work by Spengler has found very little crop by-product in hearth sediments from Begash and Tasbas. However, Spengler did report numerous rachises from mudbricks at Tasbas during the late Bronze Age phase. Ojaky in Turkmenistan also gave a number of rachises. Taken together, the frequent absence of crop by-products in the macrobotanical assemblages suggests the combined effects of taphonomy and archaeological deposition leading to poor recovery. As mentioned in the Background section of our paper, pastoralists sites are often ephemeral and dispersed, even in the IAMC up to the Iron Age, when communities began living in large villages. There however, crop processing was likely taking place off-site, as evidenced by an abundant carbonized seed record but an absence of crop by-products.

We took out "exotica" and replaced it with "goods", which we think is more neutral and straightforward term. While you and Reviewer 3 suggested to move down the text explaining that the earliest domesticated seeds in Inner Asia were recovered from mortuary contexts, we firmly believe that this information is critical for establishing the research problem that these early seed remains do not inform on subsistence or local cultivation. We greatly prefer to keep this text in the introduction, so we clarified the research problem here, which provides a better transition to the sentence introducing the dispersal model that we explore with our study.

Hordeum vulgare normally means Hulled barley, while the earliest barley in the region is naked varieties Hordeum vulgare var. nudum. So I would suggest just using Hordeum spp. or just barley to be correct.

Reply: We changed this to “*Hordeum* spp.” to represent both the hulled and naked forms that are in the macrobotanical assemblages of the sites.

In addition, I would also stay away from contracting nomadic central Asian populations with other societies living on the other ends of Eurasia as if the central Asian ones are somehow unique. Firstly, no study has really provided evidence on how mobile they really were, whether all population were mobile or just a few shepherds were moving around, and the rest are staying in winter cam all year round and growing crops. Also, there are semi mobile populations pretty much everywhere where there are domestic ovicaprids and cattle (Near East including), so central Asia in this sense is not unique.

Reply: This is a good point, and we agree that in principle there is too much conceptual baggage in contrasting Inner Asian herding communities from sedentary communities without a clearer understanding of the precise modes of mobility and labor stratification in subsistence activities. We removed “sedentary” to describe societies at either end of Eurasia who got crops that were domesticated from distant centers of domestication.

Also to be precise in the scientific publication, I would recommend to avoid using words “identical” (line 109), maybe better statistically the same; “massive migration” (line 401), how massive is massive?

Reply: We changed to “indistinguishable genetic variation”, which is a concept used by the archaeogeneticists of the cited literature. The use of the term “massive” to describe migrations in our paper is borrowed from the archaeogeneticists, which are cited. We do not agree with their characterization of the magnitude of these migrations, so we qualified it in the manuscript using “so-called”.

Unfortunately, I was not able to see the supplementary material, but hope somewhere authors explain how they calculate the percentages of millet dietary intake, this is important to explain to the reader.

Reply: Due to page limit constraints, we moved some part of the Methods section to the Supplementary Information during the initial submission. This information includes the details on calculating the percentages of C3 and C4 vegetation of dietary intake (diet-tissue spacing factors) in addition to the detailed methods for DNA sequencing and the analysis that we employed.

Few more comments:

Reference “22” is not the best here

Reply: We decided to remove this sentence, since we already establish the trans-continental spread of millet (in addition to wheat and barley) in the introduction section. We understand that there is some controversy about whether steppe pastoralists were in direct interaction with European societies, especially in Bronze Age Greece, which is what reference 22 argued.

I do not understand sentence in Line 245-247. Also please use δ next to 13C

Various water stress experiments on plants also on plants growing in higher elevations do increase the $\delta^{13}\text{C}$ values. Maybe authors have to mention this in the text in a few sentences.

Reply: The delta sign is not necessary here because the process described in this sentence explains how plants come to be characterized by higher $\delta^{13}\text{C}$ values. That is, it is due to a higher relative abundance of carbon-13 (thus, enriched in carbon-13) by recycling carbon dioxide as a result of reduced stomatal conductance. For higher $\delta^{13}\text{C}$ values in plants going up an elevational gradient, this effect is caused by increased enzymatic efficiency due to reduced partial pressure of carbon dioxide at higher elevations. We added two key citations to this sentence (Tieszen et al. 1991 and Körner et al. 1991) that are foundational for understanding the carbon isotopic variation in plants affected by water use efficiency and altitudinal effects. Because the Dzhungar mountains receive large amounts of precipitation at altitudes between 500 and 3500 m a.s.l., unlike parts of the Tian Shan in Kyrgyzstan affected by rain shadows, we have no expectation to have appreciable levels of water use efficiency in plants in mountain pastures. Likewise, higher precipitation in mountains tends to counteract the carbon-13 enrichment from partial pressure effects at altitude, per Tieszen et al. 1991.

Is it not better to use wild boar instead of wild pig in Fig. 2? Also it is not clear from the graph or the described methodology how did you make enamel compatible to collagen. Please add the sentence in methodology on it (unless it is in SOM).

In the text, in methodology is not clear if you are making the calculations on C4 dietary intake from collagen or apatite, and if from both, how do you calculate the offset values.

There is just a minor comments and the rest of methodology is clear and solid.

Reply: Yes, thank you for catching this inconsistency for the boar. We changed the figure to show “wild boar”. The methods of calculating dietary $\delta^{13}\text{C}$ values from various animal tissues (diet-tissue spacing) are described in the Supplementary Information (SI) due to page limit constraints. We think this is an appropriate place for those interested to locate these details about our methods. We added a note in the figure caption explaining where these offset values are located in the SI.

In the discussion, if you are talking about early iron age populations, the Scythians should not be the only people group that deserve mentioning. I also would ask you to clarify sentence in lines 391-392.

Reply: We added a note that these Iron Age populations are diverse and are commonly described as Scythian or Saka agro-pastoralists, and we also added “Wusun,” which we think nicely avoids the essentialist cultural groupings while still allowing readers to understand the cultural conventions for this period. We also clarified the sentence about Afanasievo groups perhaps being responsible for simultaneous or even earlier transmissions of domesticated herbivores to the Dzhungar Mountains, although our results suggest a southern source. We cannot tease apart the precise wave of transmission at this time.

Maybe the authors in the discussions could also elaborate how millet got all the way to Kazakhstan territory without the aid of pastoral societies (as ancient Chinese did not domesticate cattle or sheep/goat) and that pastoralism is not the only mean of crop translocation.

Reply: We make a specific remark about this process in the Discussion section by stating that pastoralists must have been already well-connected throughout western China and likely played an integral role in accelerating millet westward. We mention this in the context of ancient

Chinese societies not having indigenously domesticated herbivores, so millet probably moved westward slowly until pastoralist livestock were integrated into millet-based subsistence. We further emphasized this point in the text, by postulating that pastoralists may have advanced to Gansu at the same time that millet farming societies had also reached that region. After this potential contact is the time when millet likely accelerated westward.

The separate conclusion will allow to pinpoint the outstanding contribution of this article.

Reply: Thank you for this suggestion, we added a concise conclusion explaining the greater significance of the paper, especially in terms of our methodology.

Referee: 2

I find this manuscript acceptable for publication just as it is.

Reply: Thank you.

Referee: 3

1. Introduction. The opening paragraph is a little misleading in that it emphasizes the use of initial crop domesticates in mortuary contexts over the argument that is explored throughout the rest of the paper- that pastoralists spread these crops via their agricultural/economic processes. The authors might consider moving this part (lines 66-71) to the background section as part of their larger discussion of prehistoric development in the IAMC.

Reply: Thank you for this suggestion. Reviewer 1 also made the suggestion to downplay the point about ritual use of crops. Since we think this information is critical for establishing the research problem, we clarified that cannot infer local cultivation or human consumption from caches of domesticated crops deposited in mortuary contexts. Thus, having this information in the introduction provides a critical justification to explore the dispersal model of pastoralists cultivating crops and foddering animals outlined in our paper.

2. Archaeological Background. If the argument is that pastoralism precedes millet cultivation, it would make more sense to present the material chronologically and reverse the two paragraphs in this section (so paragraph 2 before paragraph 1). This rearrangement would help to establish the archaeological context of the region before introducing the specific goals of this paper. The authors might also consider similarly rearranging the introduction to help with flow and clarity (and see above).

Reply: This is a very good recommendation, and we changed the order of these paragraphs in the background section. Other modifications to the Introduction section also asked for by Reviewer 1 improve flow and clarity.

3. Materials and Methods.

a. line 126: Could you briefly elaborate on how previous faunal data clearly indicate intensive exploitation of pastoralist livestock?

Reply: We added a clause explaining that this is based on less bone fragmentation allowing for more identifications to taxon.

b. Considering the wide range of material that was analyzed for stable isotopes (collagen, apatite, enamel, hair), I would appreciate a brief explanation either in the paper or supplemental material of how or why they can be directly compared. I'm specifically thinking of Figure 2 where all of these materials are graphed using the same y-axis.

Reply: Reviewer 1 also noticed that this information was absent from the main text. We would prefer to keep the values of diet-tissue spacing in the Supplementary Information, due to the risk of running over strict page length requirements, but to help readers locate this information, we added a note to the caption of Figure 2 and also elaborated the existing note in the Methods sub-section (d).

4. Results (stable isotope analysis).

a. The cattle result are very interesting too. Did you expect them to have different isotopic signatures and/or include them as a control for detecting transhumance? Were they as important to the economy as sheep/goat?

Reply: The analyzed faunal assemblages at Tasbas and Begash show that cattle bones account for 20-30% of the total number of identifiable specimens, while sheep and goat were often >65% in all time periods. Since there are no previous studies that directly examine livestock mobility in this region, especially the differences between taxa, we did not formulate an expectation about the differences between isotopic signatures. In our original submission, we discussed the isotopic differences between sheep/goat and cattle, suggesting that cattle were managed close to settlements, while sheep/goat were likely herded farther away from the sites. A previous version of the paper had a combined Results and Discussion section. In the paper's divided sections in this version, this interesting point seemed lost. We added a small bit of interpretation in the form of a short sentence to the Results section to help the reader pickup this point later on in the Discussion, where the mobility patterns relevant to managing these animals are also better described.

b. lines 273-277: What point are you trying to make in this sentence?

Reply: The point of these sentences about cattle dietary intake is to explain that even though cattle have a smaller dietary component of C₄ plants compared to sheep and goat, the absolute mass of this fodder is substantial given that cattle have larger bodies and require more feed overall. The conceptual issue here is about moving beyond dietary percentages and helping the reader to understand the amount of C₄ biomass. We added a remark about this in relation to different strategies of foddering caprines and cattle per your previous point.

c. line 297-298: What are some examples of these "long-term agro-pastoral modifications of local ecosystems"?

Reply: We added a remark that this is likely due to the combined effects of pasturing and farming causing inputs of exogenous nitrogen to plant-soil systems. We also clarified that the livestock with low $\delta^{13}\text{C}_{\text{col}}$ and $\delta^{15}\text{N}_{\text{col}}$ values similar to wild herbivores are indicative of shared ecological niches.

d. Why does livestock consumption of millet decline as human consumption increases in mid-late 2nd millennium? And then increase again in Iron Age Begash?

Reply: In the results, we previously had a sentence about secondary dispersals of barley coinciding with a decline of millet foddering in the second millennium BC. We moved this to the Discussion at a place where readers can engage with our explanation of variable herding strategies through time (paragraph about Iron Age in Discussion). As to why millet foddering increases in the Iron Age at Begash, it is not yet possible to answer this question. One scenario could be that the initial intensification of foddering during the late third millennium BC reflects the adoption of a new food technology that pastoralists take to support their core subsistence of livestock herding. When specialized barley landraces and other crops appear in the second millennium BC, foddering strategies may have been spread out among these crops, thus effectively reducing millet fodder. By the Iron Age when millet foddering is high again, we would expect growing political and economic networks to reflect larger populations that would require more food resources. Since we cannot test either of these scenarios and since we use the Iron Age as a direct comparison of the level of production that would be possible, we would like to not dwell on the Iron Age conditions associated with rising millet foddering. However, we think your point deserved consideration, so we briefly explained our ideas about the shifts in millet cultivation between the cultural periods. Notably, this is presented in a chronological case study to walk the reader through our data in relation to the archaeological context. We also added a couple sentences explaining the decline in foddering during the second millennium BC could be due to cooler climate (e.g., d'Alpoim Guedes and Bocinsky 2018).

5. Discussion and Conclusion.

a. lines 344-348: Again, why would cattle be treated differently?

Reply: We clarified the point about cattle having been managed near settlements where there are rich water resources from mountain runoff to support the high drinking water demands of cattle. We further suggest that this could be due to a management strategy focused on dairy production, which would demand even more frequent daily watering. We also clarified that caprines were likely being herded to more arid areas than the site environs and/or to alpine pastures, which causes them to experience a dietary shift away from millet fodder to local C₃ vegetation.

b. lines 366-372: Are there other possible (non-anthropogenic) explanations for the variable nitrogen values? What is the climate like during this time period?

Reply: This is a great question. Climate records for Central Asia are poor, so considering a mechanism of climate change to account for variable nitrogen isotope values is challenging. We would expect hot and dry environments to cause soil to become enriched in nitrogen-15 due to ammonia volatilization and denitrification, which is then reflected in plant tissues. Since both caprines and cattle exhibit a wide range of $\delta^{15}\text{N}$ values, it follows that a diversity of herding practices on various areas of the landscape contributed to this pattern, some of which is likely due to exogenous nitrogen inputs from manuring and intensive pasturing. The manuring effect is visible in our data from caprines with high $\delta^{13}\text{C}$ values also having high $\delta^{15}\text{N}$ values in the absence of wild C₄ plants. On the other hand, C₃ pastures could be characterized by plants with high $\delta^{15}\text{N}$ values from pasturing or manuring C₃ crops, which we cannot disentangle for now. It is important to note here that red deer in our dataset exhibit a >4 ‰ range of $\delta^{15}\text{N}$ values, suggesting also that local nitrogen pools are variable and that these animals were ranging between highlands and lowlands. We added a remark about the natural N variation in accessible ecosystems. We also added a couple sentences explaining that climate change towards cooler conditions coincides with diversification of crops grown throughout Asia during the second millennium BC (d'Alpoim Guedes and Bocinsky 2018), and that climate records are poor for earlier periods.

c. line 380ish: Are you arguing that goats/sheep at Dali were not as transhumant as elsewhere?

Reply: Thanks for this question. We are not suggesting that Dali caprines were less transhumant than Begash caprines. We were trying to suggest that the early uptake of millet agriculture at Dali reflects the initial stages of a subsistence transformation that becomes fully fledged by the earliest occupation at Begash a few hundred years later. We added a sentence here in the paper to clarify this point, especially that the emergence of intensive millet agriculture is tied to pastoralist herding, which depends on a diversity of strategies that vary from year to year. We hope future research by us using Sr isotope analysis will resolve more precise mobility patterns and landscape use.

d. lines 383-385: What exactly does this mean? The potential links between these agropastoral practices and processes of cultural transmission should be better explained. In general this paper does a great job of interpreting results at the site/subsistence level, but doesn't fully explain the larger significance of these results or how they might articulate with other developments in this region/time period.

Reply: Thank you for pointing out this omission. We clarified the social mechanisms underlying the cultural transmission of millet through pastoralist interaction networks. This point was also emphasized by Reviewer 1, and in response, we noted that new labor divisions on account of new agricultural engagements alongside herding would also have provided a dynamic social medium for the transfer of new food technologies. We further elaborated the description of the greater significance of the study in the new conclusion section.

6. Small Details

a. line 53: Sentence is missing "of"

Fixed.

b. lines 95-99: This sentence should be rewritten for clarity

Thank you for noticing this. We made changes to the clauses of this sentence giving it better clarity.

c. line 312: Figure 4H?

Fixed. Changed to figure 3H.

d. lines 333-334: "alter" used twice in one sentence

Deleted one instance of "alter".

e. line 362: "Corresponds" or something similar instead of "resonate"

We changed to "corresponds".

f. Both BC and BCE are used at different points

We went with BC and AD, but we will change according to journal's preference.

g. The use of “agronomic” as a descriptor for economic practices throughout this paper is not one I’m familiar with (American English). A different word would be better.

We removed the term. Thank you for the recommendation.

h. Supplemental Files 4-6 should be standardized so that %C, %N, and C:N results are included in all of them.

Done, and these isotopic data were combined into one file. We also added the %C and %N data, which was mistakenly missing from the files.